# The metabolic significance of peripheral tissue clocks
A. Louise Hunter [1,2] ✉ & David A. Bechtold [1] ✉

The circadian clock is a transcriptional-translational feedback loop which oscillates in virtually all nucleated cells of the body. In the decades since its discovery, it has become evident that the molecular clockwork is inextricably linked to energy metabolism. Given the frequency with which metabolic dysfunction and clock disruption co-occur, understanding why and how clock and metabolic processes are reciprocally coupled will have important implications for supporting human health and wellbeing. Here, we discuss the relevance of molecular clock function in metabolic tissues and explore its role not only as a driver of day-night variation in gene expression, but as a key mechanism for maintaining metabolic homeostasis in the face of fluctuating energy supply and demand.

Disturbance of the circadian system, as occurs with shift work, social jetlag, and chronic sleep disturbance, has clear and established consequences for human health and wellbeing[1–6]. This includes a significant increase in risk of poor metabolic profile, insulin resistance, and type 2 diabetes (T2D). Attenuation of our body's normal rhythms and desynchrony between our lifestyle and our internal clock are recognised driving factors. Perhaps less well acknowledged is the likely impact of circadian disruption on the ability of our cells and tissues to deal with acute and chronic changes in metabolic state (e.g. via altered eating behaviour or mistimed sleep). In this Review, we consider the contribution of local circadian clock machinery in metabolic tissues such as the liver, adipose, muscle, and heart to the regulation of energy metabolic state and maintenance of normal metabolic homoeostasis in the face of challenge.

When observations are made in mammalian tissues, variation over a 24-h cycle is apparent on multiple levels: in measurements of gene and protein expression, in the activity of enzymatic processes, and in the abundance of metabolites, intermediates, and signalling molecules, for example. Sustained oscillation over repeated ~24-h cycles constitutes the internal circadian rhythm. Many such daily fluctuations are maintained even when external timing cues (such as the ambient cycle between light and dark) are removed, and laboratory animals or humans are studied under constant environmental conditions. In health, behavioural and physiological rhythms are synchronised to the external environment and coordinated across the organism, through a hierarchical system of neural, hormonal and cellular controls and signals. These signals, still not completely characterised, convey timing information from the supra-chiasmatic nucleus (SCN), the most dominant biological clock in the body, to other areas of the brain and peripheral tissues (Fig. 1). The SCN timer, which can oscillate robustly ex vivo and whose neurons can show cell-autonomous clock activity[7–9], takes its cues from the external day-night light-dark cycle via connections from the retinal-hypothalamic tract. Such circadian coordination matches an organism's biology and behaviour to anticipated daily fluctuations in the environment, and this confers a selective advantage. Upregulating processes involved in nutrient processing in advance of anticipated food availability, for example, may promote fitness by maximising the efficiency of digestion, nutrient absorption, and energy storage. Illustrating the clock's fundamental importance to health, reduced survival is observed both for wild chipmunks with SCN lesions[10] and for genetically mutant mice whose endogenous *period* (see Box 1) does not match our 24-h world[11]. In humans, when circumstances arise in which environmental cues and social/work routine misalign with internal rhythms (e.g. shift work, jet lag, experimental studies of forced desynchrony), adverse outcomes are consistently observed, particularly in metabolic traits[2,3,5,6,12]. For example, population studies demonstrate higher odds ratios for T2D[2] and obesity[3] with shift work (the odds ratio for diabetes being particularly high with rotating shift patterns[2]). In the laboratory setting, circadian misalignment reduces levels of leptin and increases postprandial circulating glucose and insulin[5].

Virtually all mammalian nucleated cells possess a molecular clock, a transcriptional-translational feedback loop (TTFL) consisting of core clock proteins, which cycles over an approximately 24-h period (Fig. 1). In addition to operating within the TTFL, these core clock proteins

[1]Centre for Biological Timing, Faculty of Biology, Medicine and Health, University of Manchester, Manchester, M13 9PT, UK. [2]Diabetes, Endocrinology & Metabolism Centre, Oxford Road Campus, Manchester University NHS Foundation Trust, Manchester, M13 9WL, UK. ✉e-mail: louise.hunter@manchester.ac.uk; david.bechtold@manchester.ac.uk

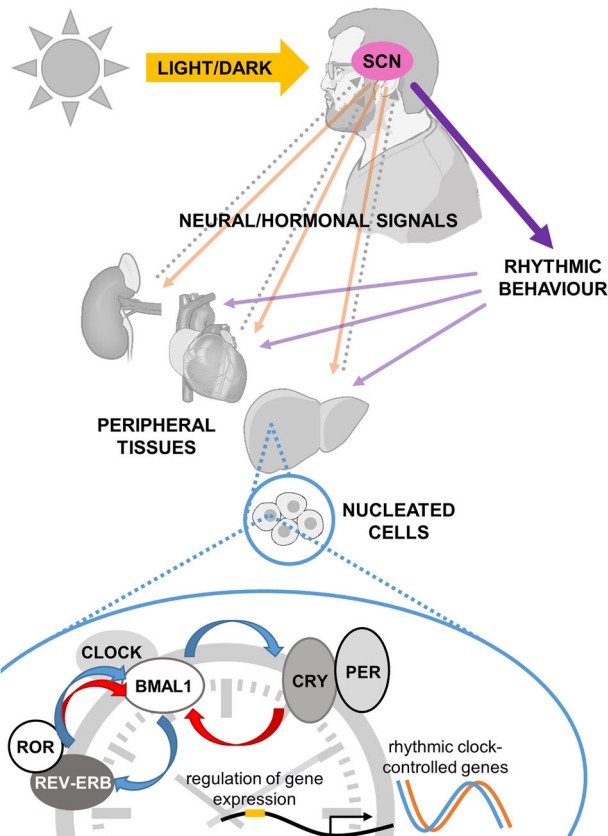

**Fig. 1 | Organisation of the circadian clock.** The suprachiasmatic nucleus (SCN) of the hypothalamus receives light/dark cues via the retinal-hypothalamic tract. The core molecular clock oscillates in nucleated cells of both the SCN and peripheral tissues (e.g. pancreas, adipose, liver, skeletal muscle, heart); this is a transcriptional-translational feedback loop (TTFL) that cycles with a 24-h period. The CLOCK:-BMAL1 heterodimer promotes the expression of repressive CRYPTOCHROME (CRY) and PERIOD (PER) proteins, which prevent further CLOCK:BMAL1-mediated transcriptional activation. CLOCK:BMAL1 also promote the expression of activator ROR and repressor REV-ERB proteins, which compete at gene regulatory elements, including those regulating core clock gene expression. Peripheral tissues are responsive to systemic signals from both the SCN and other organs and are capable of feeding back to the brain. Image created using elements from NIH BIOART[146–150].

directly regulate many thousands of genes, often imposing circadian rhythmicity onto their expression, and/or limiting their expression to certain times of day.

As we discuss in this Review, it is also clear that daily fluctuations in gene or protein expression observed in peripheral metabolic tissues cannot be attributed solely to the activity of the local clock TTFL and are driven strongly by our behaviour and multiple internal systemic signals, such as circulating hormones and metabolites. Indeed, non-intrinsic cues can be more influential in setting cell and tissue functional rhythmicity[13–15]. The core molecular clockwork can also serve to limit variances in our physiology resulting from profound daily shifts in behaviour (e.g. between sleep and active states) in processes where relative stability across day and night is required. Finally, an important, but perhaps still overlooked, function of local cell/tissue clocks is to gate cell/tissue response based on time of day. In this manner, the local clock provides the organism with a means of buffering against the effects of acute metabolic perturbation, e.g. a mistimed meal, or an extended period of fasting, whilst allowing the SCN and SCN-driven systemic signals to maintain coordination of rhythms across the organism longer-term.

## Box 1 | Circadian Terminology

*Period*—The duration of time between the start of one circadian cycle and the next.
*Phase*—When a certain point in a cycle occurs relative to a fixed point in time[25].
*Amplitude*—The difference between the maximum (or minimum) level of a variable to the midpoint level of that variable (e.g. amount of gene expression).
*Entrainment*—The process of synchronisation of an internal clock to a timing cue.
*Zeitgeber*—An extrinsic timing cue ('time giver'). For example, sunrise or when the lights of an animal house are switched on.
*Chronotype*—A person's natural preference for timing of activity or sleep[25]. For example, 'early' chronotype indicates a preference for activity in the early part of the day and, subsequently, an earlier bedtime.

## Beyond local control
### Lessons from SCN targeting

Longstanding SCN lesioning and transplantation studies highlight the predominant role of the SCN in driving rhythmicity and synchrony among behaviours and across peripheral tissues[16–18]. The SCN imposes behavioural rhythms through projections to homeostatic brain sites, which regulate behaviours such as feeding and sleep (e.g. the paraventricular and dorsomedial nuclei of the hypothalamus)[19]. In doing so, the SCN shapes autonomic, neuroendocrine, and hormonal outputs[20]. Ablation of the SCN abolishes rhythmicity of behaviour and markedly dampens peripheral tissue clock gene expression[21], whilst pairing SCN-ablated mice to intact controls by parabiosis restores rhythmic clock gene expression in liver and kidney (but interestingly, not in skeletal muscle or heart[22]). Similar recovery is observed with SCN-lesioned hamsters who receive an SCN graft, whilst liver clock genes remain arrhythmic in lesioned animals who receive a cerebral cortex graft[23]. Nevertheless, in the absence of SCN control, circadian rhythmicity can still occur. For example, in SCN-lesioned mice, the hepatocyte TTFL can oscillate in the absence of systemic cues, but at reduced *amplitude* (see Box 1), and out of synchrony with other peripheral oscillators[24–26]. Other *zeitgeber* (see Box 1) inputs can also serve to consolidate behavioural rhythms in the absence of the SCN, the most notable being regular feeding schedules and food-related *entrainment* (see Box 1). Indeed, SCN-lesioning renders peripheral tissue rhythms more susceptible to phase shifting by feeding time[25,26] (see Box 1 for the definition of *phase*); the importance of feeding as a timing cue is significant, and we discuss this in detail below.

Studies which use genetic targeting to delete or restore clock gene expression in the brain (or SCN) yield insights into the hierarchy of regulation in peripheral tissues. Mice carrying the *Clock^A19* mutation[27,28] normally show disrupted rhythms of behaviour under constant conditions, and disrupted rhythmicity in liver gene expression, compared to wild-type (WT) mice. However, restoration of *Clock* expression in the brain recovers behavioural rhythms and enhances rhythmic liver gene expression in these mice[29]. In *Bmal1* knockout (KO) mice, which do not display day-night variation in patterns of activity and food intake, recovery of rhythmic liver gene expression (compared to WT mice) is greater when *Bmal1* expression is restored to the brain, than liver[15]. Importantly, brain *Bmal1* rescue leads to some recovery of behavioural rhythms, not seen with liver rescue, again suggesting the likely role of behaviour in reinforcing rhythmicity.

### Local TTFL disruption

Many studies have taken the opposite approach: targeting the TTFL in peripheral tissues, whilst leaving the central oscillator intact. These generally

demonstrate that local TTFL activity is not essential for the rhythmicity of all oscillating transcripts, regardless of the tissue examined. One of the earliest studies arrested the liver clock through hepatocyte-targeted, tetracycline-responsive over-expression of *Rev-erbα* (*Nr1d1*), leading to repressed *Bmal1* expression[30]. This attenuated the rhythmicity of most oscillating genes assessed by liver microarray, although ~10% of transcripts continued to cycle despite the disruption of the hepatocyte TTFL. Notably, *Per2* was included in this latter group, indicating that some components of the core clock can cycle in response to external cues even when the TTFL is disrupted. This also highlights the strong connection of core clock components to internal zeitgebers, which reflect behavioural and/or metabolic states. Separate work, deleting both *Rev-erbα* and *Rev-erbβ* in hepatocytes[31], which leads to *Bmal1* derepression, but similarly arrests clock function, found a greater proportion of transcripts to be unaffected, with ~70% transcripts maintaining rhythmicity (as measured by liver RNA-seq) and ~170 genes becoming newly rhythmic in response to clock disruption. Similarly, targeting of either *Clock* or *Bmal1* in cardiomyocytes perturbed rhythmic gene expression in mouse hearts to a great extent, yet significant rhythmicity was still observed[32,33]. These studies represent only a few examples of numerous works highlighting that daily rhythms in tissue function can be maintained in the absence of local TTFL activity.

A caveat to genetic deletion studies, however, is the inability to separate outcomes resulting from altered circadian function from those resulting from loss of the clock factor per se; not trivial given that most core circadian factors are highly influential transcriptional regulators beyond action within the clockwork. The importance of this distinction is highlighted by recent works that have ablated rhythmic transcription of clock genes without deleting the genes themselves through deletion of the REV-ERB/ROR response elements (RREs) in the *Bmal1* promoter[34] and deleting the E-box in the *Per2* promoter[35]. Global deletion of the *Bmal1* RREs produces arrhythmic expression of the *Bmal1* transcript in the liver, kidney, heart and adipose but disrupts neither rhythmic expression of other clock genes in these tissues nor rhythmic circadian behaviour[34]. This is in marked contrast to transgenic mice where the *Bmal1* locus is targeted (globally) to produce a null allele[36]. Targeting of the *Per2* E-box does disrupt the rhythmic expression of wider clock genes, and produces a shortened period in mutant animals, but the circadian phenotype remains milder than that seen in *Per2* knockout mice[37]. Furthermore, some genetic manipulations (e.g. deletion of *Bmal1*, *Cry1/2*, *Rev-erbα/β* combined) have a more disruptive effect on clock cycling than others, when an individual core component of the clock can be lacking, but circadian clock function is maintained (e.g. loss of both *Rev-erbα* and *Rev-erbβ* renders fibroblasts arrhythmic in culture, whilst loss of either *Rev-erbα* or *Rev-erbβ* alone does not[38]; some redundancy between the two nuclear receptors is proposed as a possible reason for this).

## Feeding as a timing cue
### The importance of meal timing
The SCN drives daily rhythms in behaviour and, consequently, rhythms in feeding and fasting. Food intake, and especially regularly timed feeding, is a powerful timing cue for metabolic and circadian processes across many brain sites and peripheral tissues. Numerous feeding-related signals (nutrients such as glucose, fatty acids, and meal-related hormones including insulin, glucocorticoids and ghrelin) are known to act directly on the circadian mechanism[39–44]. Mice are nocturnal animals and, under *ad libitum* feeding conditions, consume the majority (~70%) of their food at night[45,46]. Restricting food access to the daytime in mice is sufficient to invert rhythms of clock gene expression in the liver, but not the SCN[47], and restoring a routine of consolidated nocturnal feeding to mice with global clock disruption (*Bmal1* or *Cry1/2* KO) restores rhythmicity to ~50% of hepatic transcripts impacted by the genetic targeting[14,48] and to a similar proportion of the circulating metabolome[15]. Remarkably, when an arrhythmic feeding regimen is imposed on WT mice, removing the rhythm of food intake, 70% of normally cycling genes lose rhythmicity of expression in the liver, despite the fact that clock gene expression, locomotor activity, and body temperature remain rhythmic[46]. Thus, loss of the (normally SCN-driven) feeding

rhythm can profoundly attenuate rhythmic hepatic gene expression, even in genetically intact animals. Whilst similar studies of peripheral gene rhythmicity in humans are relatively few, it has been demonstrated that restricting food intake to a 6-h window is sufficient to modulate clock gene expression in human peripheral blood cells[49]. Similarly, compared to an extended 16-h feeding window, 8-h time-restricted feeding leads to increased numbers of rhythmic genes in skeletal muscle (and rhythmic metabolites in serum)[50]. These findings highlight the strength of food intake as a zeitgeber to peripheral tissues, especially the liver, and its dominance over any inherent clock function.

## Mechanisms of food entrainment
The full range of mechanisms through which food intake entrains rhythms both in peripheral tissues and in behaviour are unlikely to have been fully detailed. When food availability is limited to a particular time window each day, animals rapidly display an increase in food-directed activities in advance of this window, so-called food anticipatory activity (FAA), which is a behavioural manifestation of circadian food entrainment. As noted above, clock gene expression in most peripheral and extra-SCN brain oscillators entrains to meal timing in response to restricted feeding schedules (RFS). It is proposed that behavioural control centres in the brain which exhibit circadian function become entrained to, and subsequently anticipate, repetitive mealtimes; so-called 'food-entrainable oscillators' capable of driving FAA[51]. Pathways involved in motivated and food-seeking behaviours are currently favoured candidate sites[51,52]. Although linked (and likely mutually reinforcing), it is important to recognise that the entrainment of molecular clock rhythms across the body is distinct from behavioural FAA.

Entrainment of the molecular clocks to RFS involves systemic food-related cues, such as nutrients and hormones. Interestingly, individual core clock components are differentially responsive to feeding as a timing cue. Multiple studies show persistent rhythmicity of liver *Per2* gene/protein expression, despite targeting the hepatocyte clock[13,14,30,31,48], and time-restricted feeding can restore *Per2* rhythmicity in SCN-lesioned animals[53], highlighting its responsiveness to systemic signals. Normal RFS-driven FAA is not seen in global *Per2* mutant mice[54], nor in mice with brain (or global) *Rev-erbα* deletion[55], indicating that these clock genes are central to circadian food entrainment – whereas relatively normal FAA is preserved in mice lacking other clock genes[51]. Thus, *Per2* and *Rev-erbα* appear particularly important to the coupling of circadian clock function to metabolic and feeding-related cues, and these factors clearly have an influence over energy sensing and/or energy homeostasis, which extends beyond circadian regulation.

Recent work also highlights a key contribution of insulin signalling to food entrainment across central and peripheral tissues. Insulin, which is released by the pancreas in response to carbohydrate intake, induces PER2 protein in peripheral tissues (but not the SCN) in vitro and in vivo[42], and can synchronise *Per2* expression in hypothalamic astrocytes[56]. Deletion of the insulin receptor in hepatocytes alters clock gene entrainment to feeding schedule[57], further supporting a role for hepatocyte insulin signalling in the entrainment of the liver clock. In arrhythmic feeding[46], the circadian rhythm of mTOR phosphorylation (a key component of the insulin signal transduction pathway) is lost in the livers of arrhythmically fed animals. Insulin is not the only feeding-related signal which feeds into gene regulation and the core clock. The counter-regulatory pancreatic hormone glucagon destabilises liver REV-ERBα protein, reducing REV-ERBα-mediated repression of gluconeogenesis[58]. Glucose flux influences the glycosylation of transcription factors, including BMAL1 and CLOCK, by O-linked GlcNAc transferase[39], altering the expression of their target genes. The nutrient-sensing sirtuin proteins, which are deacetylases dependent on NAD+, and therefore influenced by redox state, have been shown to influence rhythmic gene expression both at the level of the SCN[59] and liver[60]. Internal cues resulting from metabolic state can thus act as both outputs of, and inputs into the clock, reinforcing synchrony and amplitude of rhythms across tissues[20].

**Fig. 2 | Factors contributing to observed metabolic rhythms.** The SCN and central nervous system direct rhythms in behaviour (e.g. feeding), in circulating endocrine signals (e.g. glucocorticoids (GCs)), and in autonomic signalling (sympathetic and parasympathetic nervous systems (SNS, PNS)). Combined with dynamic signals derived from the environment, these systemic cues influence observed rhythms in metabolic processes.

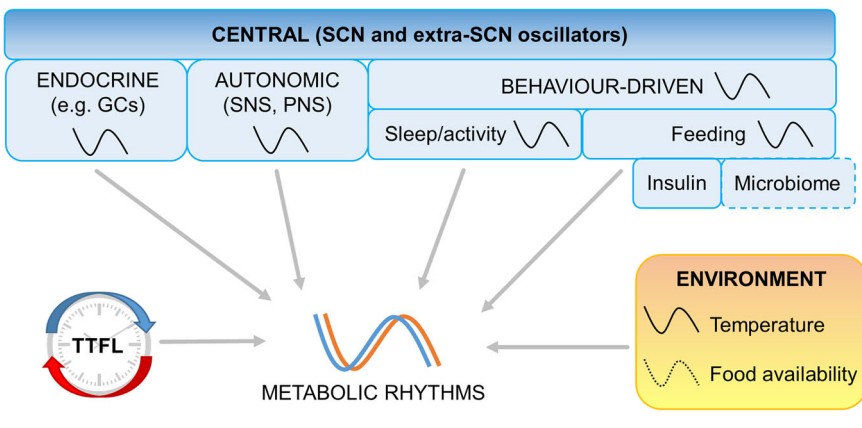

## Circadian control of feeding drive, and the role of food availability

Daily rhythms in feeding are intrinsically linked to SCN-driven rhythms in activity. That food intake is under central circadian control is supported by numerous transgenic mouse studies; mice with global or brain-targeted deletion of *Rev-erbα* exhibit disrupted patterns of food intake when kept in constant dark (DD) conditions[61,62]; mice with the *Clock^Δ19* mutation, or global *Bmal1* or *Cry1/2* KO have near arrhythmic food intake[48,63]. However, these observations likely reflect more than simply clock control of activity but also of feeding drive (or subjective hunger, in humans). Under controlled laboratory conditions, a circadian rhythm in subjective hunger is observed in humans[64,65] being lowest in the morning and highest in the evening. Mouse studies indicate that clock control of (or inherent to) the hypothalamic Agouti-related peptide (AgRP) circuitry is important here[66,67], and evidence from both human and mouse studies[65,67] suggests that the timing of hunger is informed by previous meal schedules. In nature (but perhaps less so in our modern world), there is often day–night variation in food availability and foraging pressures (e.g. risk of predation). Thus this phenomenon likely reflects the evolutionary pressures that couple food-seeking behaviour to circadian timing in order to optimise success and survival.

## Timing cues beyond food intake

Whilst food intake is arguably the strongest zeitgeber to peripheral tissues, other centrally-driven signals can also serve as timing cues to metabolic tissues. The output hormones of the hypothalamic-pituitary-adrenal axis, glucocorticoids, show robust circadian rhythmicity in rodents and humans and have potent action on rhythmic gene expression in peripheral tissues[43,68,69]. They also serve to resist the re-entrainment of peripheral clocks in response to changes in feeding time, with phase resetting of peripheral clocks occurring more rapidly in adrenalectomised animals[68]. SCN control of both sympathetic and parasympathetic activity entails that there is also a strong circadian rhythm to autonomic function[70]. Autonomic innervation of liver and adipose tissue is thought to contribute to daily rhythms of gluconeogenesis and lipolysis, respectively, with denervated tissues showing abnormal lipid/glucose handling[71,72]. Autonomic regulation of cardiac physiology is critical to heart health[73], and the interaction between circadian autonomic and cardiomyocyte clock inputs to the cardiac conducting system determines daily rhythms in heart rate, cardiac electrophysiology, and even time-of-day susceptibility to arrhythmias[74].

## Re-evaluating the role of the local clock

Taken together, it is evident that rhythmicity in peripheral metabolic tissues is the sum of multiple inputs (Fig. 2). Timing of food intake (which, in the unchallenged state, is ultimately dictated by the activity of the SCN and hypothalamic circuits), alongside neuroendocrine signalling, and contributions from other inputs, such as the microbiome[75], serves to set the phase of metabolic rhythms in peripheral tissues. This raises a question as to the role of the local clock in maintaining health.

Metabolic gene expression directly driven by the TTFL lends itself to a simple mechanistic model; one would expect to find clock transcription factor (TF) response elements in the promoters and/or enhancers of rhythmic target genes, and phase of target gene expression would correlate with the phase of the clock TF (e.g. in liver-specific *Rev-erbα* deletion, de-repressed target genes that exhibit a normal cycle with an acrophase of ZT20–22, the nadir of repressive REV-ERBα recruitment to the genome[62]). Profiling of circadian enhancer RNA (eRNA) expression in the liver illustrates that this model likely does apply to certain groups of genes[76]; eRNAs associated with amino acid metabolism peak at ZT18–ZT24, and are enriched for REV–ERB/ROR binding motifs, whilst a small number of eRNAs associated with insulin signalling peak at ZT6–ZT9, and contain E-boxes (the recognition motif for the BMAL1/CLOCK heterodimer)[76]. Clock TFs show extensive recruitment to the genome[62,77–81], implying potential for wide-ranging activity. However, many genes with clock TF binding sites do not oscillate, are not expressed, or do not cycle in phase with the TF[62,82,83], thus regulation must be more complex. Similarly, clock TF gene expression may persist unperturbed in disease states, despite gross remodelling of the rest of the rhythmic transcriptome[84–86], suggesting a pronounced change in the influence of clock TFs over gene expression.

A major role of the local clock TTFL may be to gate metabolic response according to the time of day. Such temporal gating can serve to increase the efficiency of metabolic processes and minimise futile cycling in response to regular and expected changes in metabolic and/or physiological state. *Bmal1*, for example, has been shown to be important in regulating skeletal muscle glucose utilisation[87–89] and myocardial fatty acid and glucose oxidation[33]; *Cry1/2* confers time-of-day control to muscle glycogen utilisation during exercise[90]. Similarly, clock factors may act to buffer against or minimise inappropriate or exaggerated responses to the same signals when received at the wrong time of day (e.g. a midnight snack). This influence is likely achieved through TTFL-driven rhythms in the expression of relevant cellular components (e.g. key enzymes and cell surface receptors), but also TTFL regulation of acute transcriptional response to incoming signals. Here, clock control is exerted by direct repression at gene sites (e.g. by core clock REV–ERB and CRY proteins) and through clock regulation of differential chromatin accessibility across the day and night (Fig. 3). In this way, disruption of the local clock machinery or its components increases susceptibility to metabolic pathology.

## Local clocks in the maintenance of homoeostasis and adaptive metabolism

### Metabolism in clock-targeted mice

Transgenic mice in which the functioning of the circadian clock has been disrupted commonly exhibit abnormal metabolic phenotypes. *Clock* mutant mice demonstrate obesity and hyperglycaemia[63], mice with global deletion of *Rev-erbα* show abnormal lipid and carbohydrate handling[91,92], and greatly impaired skeletal muscle exercise capacity[93], global deletion of *Bmal1* is associated with adiposity and altered liver glucose output[94–97]. As discussed,

**Fig. 3 | A model of local clock action.** The role of the tissue clock is to gate the activity of rhythmic physiological processes, which are ultimately driven by central cues. This may be through suppressing or amplifying the influence of behaviour-driven systemic signals. This may arise through modulation of key metabolic regulators (e.g. PDK4, which phosphorylates pyruvate dehydrogenase), or through regulation of the transcriptional response to metabolic state change. Image created using an element from NIH BIOART[151].

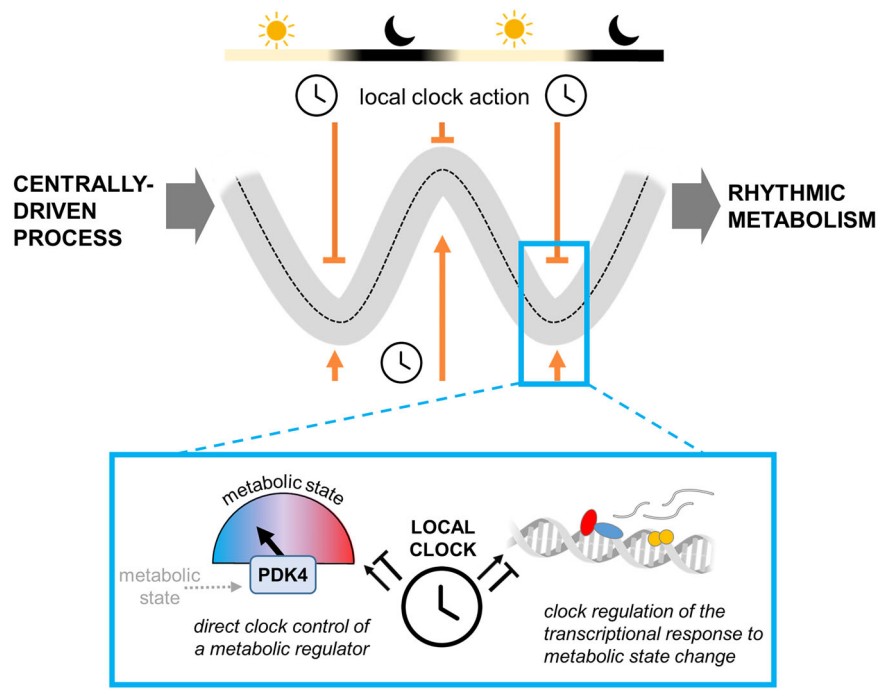

core clock transcription factors have extensive genome-wide binding activity in the liver[62,77,79,80], adipose[78], heart[83,98] and muscle[81], which implies a broad scope of control.

However, findings from tissue-targeted models do not often replicate the phenotypes observed in globally-targeted animals. Pancreas-targeted *Bmal1* deletion results in impaired glucose-stimulated insulin release and persistent hyperglycaemia across the day–night cycle[99], a phenotype distinct from that of globally targeted mice. Targeting *Bmal1* deletion to skeletal muscle has effects on whole body composition and glucose handling[87,89], but the muscle phenotype is milder than with global *Bmal1* KO. Mice with hepatocyte-targeted *Bmal1* deletion have normal circadian activity and feeding profiles[96], but have lower circulating glucose levels in the resting (fasting) state, and lower liver glycogen levels, suggesting that liver *Bmal1* helps to maintain hepatic glucose output (proposed to be through regulation of glucose export by GLUT2), and thus constant blood glucose levels[96,100]. In our own work, mice with liver- or adipose-targeted deletion of *Rev-erbα*, exhibit minimal impact on metabolic phenotype in the basal state[62,78], which implies that the role of REV–ERBα is not simply a *de facto* regulator of lipid metabolism, as suggested by some previous findings[38,79,91,101].

Such examples highlight the challenges in dissecting the influence of the local clock in maintaining metabolic homoeostasis and in disentangling critical points of influence. Nevertheless, some clear mechanistic targets have emerged, exemplifying how and where TTFL control over critical enzymes and regulators can exert profound control over energy state or adaptive metabolic response (Fig. 3). For example, CRY has a negative effect on phosphorylation (and thus activation) of cAMP response element–binding protein (CREB), a key regulator of gluconeogenesis[102]. Hepatic over-expression of *Cry1* thus lowers fasting blood glucose levels in mice[102]. PDK4, which phosphorylates (inactivates) pyruvate dehydrogenase (PDH), is strongly circadian in several tissues, including liver[103] and skeletal muscle[87,104], and is disrupted by muscle *Bmal1* knockout[87]. PDH converts pyruvate to acetyl-CoA, so directing the products of glucose oxidation into the TCA cycle, and mice with muscle *Bmal1* deletion show both reduced PDH activity and reduced rates of glucose oxidation[87]. TTFL-driven rhythmicity in the activity of these critical factors thus permits time-of-day control over metabolic pathway flux.

## Multifactorial direction of rhythmic gene expression

By returning to those studies which have employed genetic targeting to isolate peripherals from central oscillators, we can gain further insights into the functions of local clocks. Specifically, when the hepatocyte clock is targeted through *Bmal1* deletion, *Rev-erbα* over-expression, or *Rev-erbα/β* deletion (but behavioural and feeding rhythms remain unaffected), a number of transcripts lose rhythmicity[30,31,96] while others gain rhythmicity[31]. In the study of *Rev-erbα/β* deletion[31], those genes which lost rhythmicity were enriched for circadian clock, hormone secretion, and lipid metabolism pathways and associated with binding sites for core clock transcription factors (REV-ERBs, CRY1, PER2) and liver lineage determining factors (HNF4A, FOXA2). Those genes which gained rhythmicity were enriched for lipid metabolism pathways and associated with SREBF1 binding sites, a sterol-responsive transcription factor known to regulate lipid synthesis. In a mouse model with *Bmal1* expression limited to the liver, despite the arrhythmic feeding and locomotor activity of these mice, a small number of liver metabolites and transcripts oscillate, notably those participating in carbohydrate (glycogen) and redox (NAD+) metabolic pathways[13]. Other transcriptomic studies[14,46,48] have combined altered feeding regimens with clock targeting in order to systematically group genes by their responsiveness to local clock control and meal timing. Whilst strategies differ and resulting gene clusters vary, the conclusions are largely similar: there are transcripts which cycle independently of the clock and of feeding pattern, perhaps responsive to other systemic signals (e.g. glucocorticoids), transcripts whose cycling is dependent on the clock (including the core clock genes), transcripts driven predominantly by feeding rhythm, and transcripts whose proper cycling requires both the local clock and a feeding rhythm. In all these analyses, transcripts entirely dependent on the local clock are in the minority.

The finding that larger numbers of genes are regulated jointly by the clock and systemic signals implies that there is a role for local clocks in interpreting these systemic signals, for example in amplifying the transcriptional response to a rhythmic systemic input. Additionally, those studies in which rhythms newly emerge in clock-targeted peripheral tissues[15,31,105] reinforce the notion that the local clock may be important in suppressing undesirable rhythmicity. In the context of energy metabolism, such amplification and suppression is likely an important component of

energy homoeostasis in the face of fluctuating energy supply and demand (Fig. 3).

## A role for the clock in response to challenge

The clock's importance to adaptive metabolic response and energy homoeostasis is further revealed by observations showing that the phenotypic impact of deleting clock components is frequently exacerbated or unmasked under acute or chronic metabolic challenges (e.g. fasting, mistimed nutrient load, obesity). We have found this with mice with liver- or adipocyte-targeted deletion of *Rev-erbα*, which show only minor metabolic alterations under normal feeding conditions, but where a much more profound role of REV-ERBα is revealed by mistimed feeding or high fat diet[62,78]. *Per2* mutant mice (*Per2^Brdm1^*) have normal blood glucose control in the fed state but become abnormally hypoglycaemic upon daytime fasting, and show reduced ability to synthesise glycogen upon refeeding[106]. Loss of cryptochromes *Cry1* and *Cry2* renders mice less able to return to euglycaemia following a glucose load[107], and increases glucose output by primary hepatocytes[102], whilst *Cry1* over-expression improves handling of both glucose and pyruvate loads[102]. Acute treatment with CRY activator KL001 affects gluconeogenic gene expression when hepatocytes are stimulated with glucagon, but not in the basal state[108]. These findings are compatible with the local tissue clock acting as a buffer against the impact of acute perturbations in the metabolic state. In the short term, this buffering is beneficial, as metabolic homoeostasis is maintained, across the day-night, feeding-fasting schedule, and in the face of any acute changes in nutrient availability (e.g. acute scarcity of food, acute abundance of food). It would be detrimental for a single mistimed meal to elicit an improper systemic response or to shift rhythmic activity, thus resulting in misalignment over the proceeding days. Of course, as 'mistimed' feeding becomes established, as in experimental restricted feeding schedules, re-entrainment of the clock facilitates anticipatory and appropriate responses.

Studies have also examined the role of the local clock in the adaptation to chronic metabolic challenges, e.g. persistent hypercalorific feeding and resultant diet-induced obesity. These show mixed effects of clock deletion in the face of chronic metabolic perturbation. For example, with adipocyte-targeted *Rev-erbα* deletion, enhanced adiposity, preserved insulin sensitivity, and reduced ectopic lipid storage are observed when mice are fed long-term HFD[78]. Thus, in the case of *Rev-erbα*, it appears that this clock factor serves to regulate adipocyte expansion and lipid storage, but in a way that proves deleterious with chronic lipid excess—a state unlikely to have been experienced in mammalian evolution. In chronic high-fat diet feeding conditions, the limiting of adipose fat storage occurs at the expense of increased ectopic fat deposition, increased hepatic glucose output, and adipose tissue dysfunction, so predisposing to metabolic disease. Similarly, the lipogenic effects of liver *Rora/Rorc* deletion are unmasked with HFD feeding, with targeted animals showing significantly higher liver triglycerides and greater numbers of differentially expressed genes than are apparent in the normal chow (NC) fed state[109]. ROR activator nobiletin does not affect metabolic phenotype in NC-fed mice but does protect against obesity when mice are challenged with HFD[110]. The impact of adipocyte *Bmal1* deletion is more apparent with HFD feeding[111]. *Cry1* KO mice have a similar body composition to WT mice when fed a normal diet, but show reduced gain in fat mass when fed HFD for 16 weeks[112]. Whilst results vary, the common finding in these studies is that the importance of the local clock function becomes more apparent when the metabolic state is chronically perturbed.

## A model for local clock action

In sum, these observations lead to a model (Fig. 3) in which a key role of the local clock is to maintain energetic homoeostasis, both in the face of day-night variation in nutrient supply and demand and by buffering against the effects of acute metabolic perturbation. This is achieved in part through rhythmically gating the function of key enzymes (e.g. PDK4), but perhaps largely through gating (both amplifying and repressing) the transcriptional response to systemic signals and changes in metabolic state. We can

hypothesise that the responsiveness of certain TTFL components to feeding signals and metabolic state may further promote acute and long-term metabolic adaptation[113]. Phosphorylation and, therefore, destabilisation of PERIOD and CRY proteins mediated by metabolic sensor AMPK[102,114,115] (either via phosphorylation of casein kinases, or even directly by AMPK itself) is one such putative mechanism. The focus of this review has been the local TTFL clock, but it should also be mentioned that cell-autonomous circadian rhythms are observed in enucleated cells lacking a TTFL, such as red blood cells[116,117]. As the underlying mechanisms and consequences of such rhythms are elucidated, this should inform our understanding of their importance in nucleated cells and how the local clock interacts with rhythms in redox states and in post-translational events more broadly.

## Outlook

### The importance of a local clock for metabolic health

At the level of transcriptional regulation, TFs do not function in isolation, and gene regulation is achieved by combinatorial transcription factor activity, and modulation of the chromatin environment[118–120]. At each of these points, the context of TF activity can be modified by energy state, in a cell-type specific manner. Clock TFs can physically interact with other key regulators, including nuclear hormone receptors[121] such as the PPARs and GR. The chromatin state is altered by histone modifications such as lysine acetylation, which can be modulated by changes in cellular acetyl-CoA or NADPH[122,123]. Thus, the expression pattern of metabolic genes reflects multiple variables. By influencing, and being influenced by, these transcriptional and chromatin dynamics, the local clock sets the regulatory context within which cellular and tissue responses to external signals and changes in metabolic state occur.

Considering this wider perspective of local clock action (Fig. 3), why might metabolic pathology be observed so frequently in studies of shift work and other forms of circadian disruption[2,3,5,6,12]? When there is a mismatch between behaviour and internal rhythms, input to the system (e.g. a meal, a bout of exertion) comes at a time when the response of the organism is inappropriate, or blunted (e.g. reduced insulin sensitivity is seen in humans during a forced desynchrony protocol[6]). Mistimed meals will result in irregular input to the peripheral tissue clock, especially to metabolically-sensitive components of the TTFL such as *Per2* and *Rev-erbα*, and misalignment between behaviour (i.e. meals, sleep/activity cycles) and oscillation of the TTFL will alter the metabolic environment within which the TTFL operates. Combined, the result may be both damping of desirable rhythms in metabolic processes, a failure of the TTFL to suppress undesirable rhythms in others (e.g. an increase in lipogenesis during inactivity), and a mismatch between substrate availability and substrate processing, impairing energy efficiency[124]. In the heart, for example, a highly metabolically active organ which undergoes substantial shifts in substrate demand and availability over the day–night cycle, it is easy to see how circadian disruption could trigger pathology[69,125].

### Understanding and addressing local clock disruption in humans

There is a need to determine how we can best mitigate the disruption of local tissue clocks in those people whose sleep–wake cycle and behavioural rhythms are misaligned with the external environment. Such misalignment between behavioural rhythms and the external environment in humans commonly occurs through shift work, social jetlag, chronic sleep disruption, and extremes in *chronotype* (see Box 1). As highlighted, there is a strong evidence base linking shift work with metabolic ill health[1–6], including obesity and T2D risk. Individuals with, for example, an extreme late chronotype, may rise late in the day and stay awake long into the hours of darkness. Analysis of UK Biobank data has linked greater 'eveningness' to increased T2D risk and increased all-cause and cardiovascular disease mortality[126]. Other studies have found associations between chronotype and variables as diverse as vascular endothelial function[127], lipoprotein levels[128], insulin sensitivity[129], body mass index, and food preferences[130], with, interestingly, favourable metabolic profiles always associating with early chronotype.

Aside from chronotype, researchers have investigated the importance of meal timing for metabolic disease risk in humans. An analysis of NHANES data found that 59% of American adults they studied took in calories after 9 p.m. at night[131]; thus, late mealtimes are a common occurrence. Some studies have reported an association between later mealtimes and increased adiposity/obesity risk[132,133], but the evidence base is mixed[134], and there are many confounding factors which have to be taken into account.

Based on both these observations in humans and what has been learnt from animal models, time-restricted feeding has attracted considerable interest. Here, mealtimes are limited to a specific time window (e.g. an 8-h window during daytime hours), but without an intended limitation in calorie intake. This is intended to improve the regularity of feeding as a cue to peripheral tissues and reduce misalignment between mealtimes and the peripheral clock. Food restriction to an 8-h night-time window was first shown to be effective at improving whole-animal metabolic parameters in mice[135], and similar time-restriction regimens have since been shown, in multiple small studies, to be of metabolic benefit in humans[134,136–139]. A recent feasibility study of 137 people (firefighters) working 24-h shifts found no adverse effects and reported some metabolic improvements (more so in people with existing cardiometabolic risk) in those who limited mealtimes to a 10-h window[140]. However, there are also studies which have shown an absence of benefit[141,142], including a study of 100 participants in which change in body weight was the primary outcome[142].

Whether or not time-restricted feeding is a feasible intervention for clinical benefit will become clearer as current and ongoing studies are completed and interrogated. Furthermore, whilst accumulating evidence links human clock gene polymorphisms to metabolic traits[143–145], highlighting the importance of the TTFL to metabolic health, we do not yet know how to exploit this information for therapeutic purposes. Multiomic cell atlases, which are starting to emerge, have promise as a tool to help determine the tissue(s) in which these polymorphisms are most influential.

## Conclusion

In summary, it is evident that an intact local clock is necessary for optimal metabolic health. To thrive in a fluctuating metabolic environment, we need the local clock to maintain homoeostasis, gating the response to systemic cues and buffering against the impact of varying meal times. This hypothesis is difficult to test, but the clock and metabolic dysfunction that is seen with chronic circadian disturbance are strongly supportive. Recognising that local clocks make a key contribution to metabolic health means that their function should be considered when studying metabolic pathology in shift workers and when designing measures to mitigate such pathology.

### Reporting summary

Further information on research design is available in the Nature Portfolio Reporting Summary linked to this article.

### Data availability

This review article did not analyse or generate any new datasets.

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

## Acknowledgements

Funding: A.L.H. is supported by the Wellcome Trust [225145/Z/22/Z].

## Author contributions

A.L.H.: Writing—original draft, Writing—review & editing; D.A.B.: Writing—review & editing.

## Competing interests

The authors declare no competing interests.
