## [Transparent Peer Review file · Communications Biology]

The metabolic significance of peripheral tissue clocks

Corresponding Author: Dr A. Louise Hunter

Version 0:

Reviewer comments:

Reviewer #1

(Remarks to the Author)

Hunter et al have reviewed the metabolic significance of peripheral tissue clocks. This is a detailed review which summarizes the working of the circadian clocks, both central, and peripheral, along with a summary of basic and animal experiments related to food timing, metabolic mechanisms and their relation to peripheral clocks. The authors propose a model in which the main role of the local clock is to maintain energetic homeostasis, while balancing the day-night variation in nutrient supply and demand, and by buffering against the effects of acute metabolic perturbation.

I have a few suggestions:

The authors should consider numerous sub headings as currently this manuscript is very dense to read. For instance, the section "Local clocks in maintenance of homeostasis and adaptive metabolism" needs to be broken down into several parts either by organ system, or experimental set up.

In the proposed model, the concept of a day-night variation in nutrient supply needs to be clearer. For instance, limited food supply was an evolutionary problem that is somewhat limited now. At least in the developed world, the problem is availability of high energy food at all times of the day. Limited food availability may be still present in the animal world, but since these summaries will eventually benefit humans, the concept of day-night variation in food supply should be revisited. Additionally, in Figure 2, there is a rhythm in food availability (yellow box); this should be reconsidered.

At various places, there is a discussion of the amplitude of rhythms. How animal/basic experiments also alter the phase of should also be explained better, along with what reduced amplitude/phase mean.

A separate section on human work would benefit the readership. Understandably, there is not much work in peripheral clocks in humans, but the authors mention time restricted eating and cite a few human studies of circadian misalignment. The human section could also involve timing of food and obesity, and roles of chronotype.

Specific concerns, line 215: "Timing of food intake (which is ultimately dictated by the activity of the SCN)", is a board statement, and this may not be only limited to the activity of the SCN (e.g., states in which glycogen repletion is necessary). Below, line 218: 'under normal circumstances', is vague. Do the authors mean in health?

Reviewer #2

(Remarks to the Author)

Comments for Authors

In this paper, Hunter et al. describe the links between circadian clocks, metabolism and global health. More specifically, the authors highlight the importance of peripheral clocks, how they work together with the central clock, integrate external signals and facilitate metabolic tissue homeostasis in metabolically active organs. This is a very well-written comprehensive review article on a timely topic, approaching the subject from an interesting angle.

Here, I list some points which might improve the manuscript:

Major:

-This review highlights important metabolic organs, but predominantly focuses on liver, adipose, and skeletal muscle tissues. Since the heart is a highly metabolic organ and clock protein disruption (tissue-specific, as well as whole body KOs) results in severe metabolic phenotypes, it might be nice if a more in-depth description of this organ could be implemented in this review article.

Minor:

-The authors stress the presence of clocks in nucleated cells, but could also mention circadian oscillations in enucleated cells, such as terminally differentiated red blood cells. Yet, they do not display a TTFL.

-Line 75: References to the Bmal1:RE models should be inserted.

-Line 122: The authors describe that the outcome of genetic clock factor knockout experiments cannot unequivocally be attributed to altered rhythms or loss of the factor per se. Abe et al. (PMID: 35999195) generated Bmal1 RRE mutants and found that circadian molecular oscillations were normal. In addition, Doi et al (PMID: 31189882) constructed Per2 E-box mutants that display impaired circadian oscillations. The authors could mention these two models, and compare their phenotypes with Bmal1 and Per2 whole body knockouts. Comparison should teach us something about whether phenotypes are mediated by the lack of circadian rhythmicity or the factors per se.

-Line 260: The authors mention that clock factors have extensive genome-wide binding activity in liver, adipose tissue and muscle, but could also mention the heart: PMID: 35036997 and PMID: 30804225.

-Line 330: the phenotypic impact of clock gene deletion is often unmasked upon challenge (e.g. HFD leads to aggravated fatty livers in Rev-erb α KO animals). Is the consensus that under these kind of stimuli (in WT animals) TFs reposition (bind to separate sites) and regulate different genes, or bind to the same sites, but other regulators contribute to gene (de)regulation?

Version 1:

Reviewer comments:

Reviewer #1

(Remarks to the Author)

The authors have answered my comments satisfactorily.

Reviewer #2

(Remarks to the Author)

The authors have addressed all my issues and implemented my suggestions. I have no further concerns.

Response to Reviewers' Comments:

Reviewer #1 (Remarks to the Author):

Hunter et al have reviewed the metabolic significance of peripheral tissue clocks. This is a detailed review which summarizes the working of the circadian clocks, both central, and peripheral, along with a summary of basic and animal experiments related to food timing, metabolic mechanisms and their relation to peripheral clocks. The authors propose a model in which the main role of the local clock is to maintain energetic homeostasis, while balancing the day-night variation in nutrient supply and demand, and by buffering against the effects of acute metabolic perturbation.

I have a few suggestions:

- 1) The authors should consider numerous sub headings as currently this manuscript is very dense to read. For instance, the section “Local clocks in maintenance of homeostasis and adaptive metabolism” needs to be broken down into several parts either by organ system, or experimental set up.

Thank you – we have now inserted numbered subheadings throughout the main body of the text. We hope the reviewer agrees that readability is improved. These subheadings, along with other text amendments addressing Referee comments are specifically marked in blue on the manuscript.

- 2) In the proposed model, the concept of a day-night variation in nutrient supply needs to be clearer. For instance, limited food supply was an evolutionary problem that is somewhat limited now. At least in the developed world, the problem is availability of high energy food at all times of the day. Limited food availability may be still present in the animal world, but since these summaries will eventually benefit humans, the concept of day-night variation in food supply should be revisited. Additionally, in Figure 2, there is a rhythm in food availability (yellow box); this should be reconsidered.

This is an important point. To take this into consideration, we have added a section on clock control of feeding drive (hunger) (section 3.3., lines 229-243), and discussed how day-night variation in nutrient supply (or lack thereof) may relate to this. We have made a slight adjustment to Figure 2 (yellow box) to acknowledge this point too.

- 3) At various places, there is a discussion of the amplitude of rhythms. How animal/basic experiments also alter the phase of should also be explained better, along with what reduced amplitude/phase mean.

Thank you for this suggestion. We have added a Box (Box 1) to provide definitions of circadian terminology, such as ‘amplitude’ and ‘phase’, for the reader, plus have added to the text at lines 108-110 to describe how SCN lesioning affects circadian phase.

- 4) A separate section on human work would benefit the readership. Understandably, there is not much work in peripheral clocks in humans, but the authors mention time restricted eating and cite a few human studies of circadian misalignment. The human section could also involve timing of food and obesity, and roles of chronotype.

We agree that this would strengthen our article. We have therefore substantially expanded the section on human work in the Discussion (section 5.2., lines 464-503), incorporating the topics suggested.

- 5) Specific concerns, line 215: “Timing of food intake (which is ultimately dictated by the activity of the SCN)”, is a broad statement, and this may not be only limited to the activity of the SCN (e.g., states in which glycogen repletion is necessary).

Thank you for this observation – we have changed the wording of this line (now line 267) to try and convey our meaning more clearly. We acknowledge that in states of metabolic challenge (e.g. food withdrawal leading to glycogen depletion), food intake may be directed by other neuroendocrine circuits.

- 6) Below, line 218: ‘under normal circumstances’, is vague. Do the authors mean in health?

Yes, ‘in health’ is indeed our intended meaning! We have amended this sentence accordingly (line 270-271).

Reviewer #2 (Remarks to the Author):

In this paper, Hunter et al. describe the links between circadian clocks, metabolism and global health. More specifically, the authors highlight the importance of peripheral clocks, how they work together with the central clock, integrate external signals and facilitate metabolic tissue homeostasis in metabolically active organs. This is a very well-written comprehensive review article on a timely topic, approaching the subject from an interesting angle.

Here, I list some points which might improve the manuscript:

Major:

- 1) This review highlights important metabolic organs, but predominantly focuses on liver, adipose, and skeletal muscle tissues. Since the heart is a highly metabolic organ and clock protein disruption (tissue-specific, as well as whole body KOs) results in severe metabolic phenotypes, it might be nice if a more in-depth description of this organ could be implemented in this review article.

Thank you for this suggestion. The heart is indeed an important metabolic organ, and we have updated our review to include greater mention and discussion of heart studies, and what we can learn from these. Specifically, we have added to the text at lines 137-139, lines 255-258, line 317, lines 460-462.

Minor:

- 2) The authors stress the presence of clocks in nucleated cells, but could also mention circadian oscillations in enucleated cells, such as terminally differentiated red blood cells. Yet, they do not display a TTFL.

It is certainly important to consider what we can learn from cells lacking a local TTFL, and we have added to the text to include this point (lines 429-434).

- 3) Line 75: References to the Bmal1:RE models should be inserted.

Thank you for this helpful suggestion. We have inserted the appropriate references (now line 81).

- 4) Line 122: The authors describe that the outcome of genetic clock factor knockout experiments cannot unequivocally be attributed to altered rhythms or loss of the factor per se. Abe et al. (PMID: 35999195) generated Bmal1 RRE mutants and found that circadian molecular oscillations were normal. In addition, Doi et al (PMID: 31189882) constructed Per2 E-box mutants that display impaired circadian oscillations. The authors could mention these two models, and compare their phenotypes with Bmal1 and Per2 whole body knockouts. Comparison should teach us something about whether phenotypes are mediated by the lack of circadian rhythmicity or the factors per se.

Thank you, these are important papers, and we now include them in our Review, and discuss what we can learn from their findings (lines 146-154). It would be very interesting to compare, specifically, the metabolic phenotypes of these models with the global Bmal1 and Per2 knockouts, but we weren't able to find this data published.

- 5) Line 260: The authors mention that clock factors have extensive genome-wide binding activity in liver, adipose tissue and muscle, but could also mention the heart: PMID: 35036997 and PMID: 30804225.

Thank you, we have amended the text accordingly (now line 317).

- 6) Line 330: the phenotypic impact of clock gene deletion is often unmasked upon challenge (e.g. HFD leads to aggravated fatty livers in Rev-erba KO animals). Is the consensus that under these kind of stimuli (in WT animals) TFs reposition (bind to separate sites) and regulate different genes, or bind to the same sites, but other regulators contribute to gene (de)regulation?

This is a fascinating question, which we have also previously debated! There is arguably evidence for both phenomena, if we extrapolate from studies of non-clock TFs. For example, a recent study found evidence for TF (AP-1) repositioning over the course of the ageing process (PMID: 38959897) due to changes in chromatin accessibility. But there is also evidence for altered co-occupancy by other regulators affecting TF-mediated gene expression (e.g. STAT5 and GR; PMID: 31706703). Whilst we feel that the gene regulatory environment is beyond the scope of this Review, we have adjusted text in our Discussion (line 445) to reflect this point.